# VHL L169P Variant Does Not Alter Cellular Hypoxia Tension in Clear Cell Renal Cell Carcinoma

**DOI:** 10.3390/ijms241814075

**Published:** 2023-09-14

**Authors:** Junhui Hu, Desmond J. Smith, Lily Wu

**Affiliations:** 1Molecular and Medical Pharmacology, University of California, Los Angeles, CA 90095, USA; dsmith@mednet.ucla.edu; 2Institute of Urologic Oncology, University of California Los Angeles, Los Angeles, CA 90095, USA; 3Brain Research Institute, University of California Los Angeles, Los Angeles, CA 90095, USA; 4Jonsson Comprehensive Cancer Center, University of California Los Angeles, Los Angeles, CA 90095, USA

**Keywords:** *VHL*, clear cell renal cell carcinoma, ccRCC, L169P, T506C, hypoxia, protein degradation

## Abstract

In the current era of tumor genome sequencing, single amino acid missense variants in the von Hippel–Lindau (*VHL*) tumor suppressor gene are frequently identified in clear cell renal carcinoma (ccRCC). Due to the incomplete knowledge of the structural architecture of *VHL* protein, the functional significance of many missense mutations cannot be assigned. L169P is one such missense mutation identified in the case of aggressive, metastatic ccRCC. Here, we characterized the biochemical activity, transcriptomic hypoxia signature and biological functions of the L169P variant. Lentiviral vector expressing either wildtype (WT) or L169P *VHL* were used to transduce two *VHL*-deficient human ccRCC cell lines, 786-O and RCC4. The stability of the *VHL* protein and the expression level of *VHL*, *HIF1α* and *HIF2α* were analyzed. The impact of restoring L169P or WT *VHL* on the hypoxia gene expression program in 786-O cells was assessed by mRNA sequencing (RNAseq) and computed hypoxic scores. The impact of restoring *VHL* expression on the growth of ccRCC models was assessed in cell cultures and in chorioallantoic membrane (CAM) xenografts. In the 786-O cells, the protein stability of L169P *VHL* was comparable to WT *VHL*. No obvious difference in the capability of degrading *HIF1α* and *HIF2α* was observed between WT and L169P *VHL* in the 786-O or RCC4 cells. The hypoxic scores were not significantly different in the 786-O cells expressing either wildtype or L169P *VHL*. From the cellular function perspective, both WT and L169P *VHL* slowed cell proliferation in vitro and in vivo. The L169P *VHL* variant is comparable to WT *VHL* in terms of protein stability, ability to degrade *HIF1α* factors and ability to regulate hypoxia gene expression, as well as in the suppression of ccRCC tumor cell growth. Taken together, our data indicate that the L169P *VHL* variant alone is unlikely to drive the oncogenesis of sporadic ccRCC.

## 1. Introduction

The *VHL* (von Hippel–Lindau) tumor suppressor gene is the most frequently mutated gene in clear cell renal cell carcinoma (ccRCC) followed by *PBRM1*, *BAP1* and *SETD2*. Alterations in the *VHL* gene, such as missense mutations, deletion or epigenetic silencing, have been identified in all familial cases and about 85% of sporadic cases of ccRCC [1]. Since the recognition of “*VHL* disease” in 1936, over 1600 mutations have been identified in this gene from ccRCC tumor tissues [2,3]. Amongst the identified *VHL* mutations, over 900 are missense mutations with a single amino acid substitution [3]. Most of these missense mutations were evaluated by computation models without comprehensive characterization in cellular or animal models [4]. At this time, an atlas that accurately assigns a *VHL* genotype to its oncogenic phenotypes in ccRCC is lacking. *VHL* L169P is one such missense variant of unknown significance in exon 3, identified in a highly aggressive and metastatic case of ccRCC [5] and in four patients in the TCGA-KIRC and CPTAC-3 projects [6].

Three isoforms of *hVHL* have been identified, including the 213 aa (amino acids) isoform 1 (*hVHL*-v1), the 172 aa isoform 2 (*hVHL*-v2) that skips the second exon in hVHL-v1, as well as the third isoform (hVHL-v3) with 193 aa that has a distinct C-terminal exon from the other two isoforms. In addition, *hVHL*-v1 can be translated into two protein products due to an alternative start codon at amino acid 54, resulting in a 30 kD and a 19 kD protein [7]. The protein structure of hVHL is composed of a β-domain in the N-terminal and an α domain in the C-terminal [8]. The L169P substitution impacts the *hVHL*-v1 and v2 structure at the C-terminal α-domain.

*VHL* protein is known to be involved in two major cellular processes. First, it is well recognized as a component of an E3 ubiquitin ligase VCB–CR complex (**V**hl/Elongin **C**/Elongin **B**–**C**ullin 2/**R**ING finger protein), which is responsible for binding and degrading hypoxia-inducible factors (*HIF1α* and *HIF2α*) under normoxic condition [8,9,10]. The N-terminal β-domain participates in substrate recognition and binding to *HIFαs*, while the α-domain interacts with elongin B and C [8,11,12]. Therefore, mutations in this domain including L169P mainly affect the interface between pVHL and elongin C/B and are classified as type 2b *VHL* mutations [13,14,15]. The second known function of *VHL* is involved in regulating cilia centrosome and microtubules to maintain cell polarity and chromosomal stability in cell division [16,17].

The *VHL* function in the first process, namely the regulation of the hypoxic response [18], is considered to be the main cause of ccRCC tumorigenesis. A wide range of *VHL* protein dysfunctions, ranging from the complete loss of *VHL* protein to the disabling of its enzymatic function, or just a reduced protein stability resulting in protein insufficiency, could contribute to oncogenesis attributed by dysregulated hypoxic responses. In this study, we undertook a comprehensive functional evaluation of the L169P *VHL* variant regarding its protein stability, cellular hypoxia tension and tumorigenic potential in two human ccRCC models. 

## 2. Results

### 2.1. Amino Acid L169P Substitution Does Not Alter VHL Protein Stability

Over the past 5 years, we collected fresh surgical tumor samples from 50 patients with RCC and generated 40 primary cell lines from these samples to study their biology [5]. The most aggressive case amongst this cohort was a 59-year-old male who presented with a 10 cm, Fuhrman grade 4, primary tumor and bilateral lung metastases and who succumbed within one year of nephrectomy despite multiple interventions. The L169P *VHL* variant was identified in the tumor of this aggressive case of ccRCC by whole-exome sequencing (WES), revealing a homozygous in-frame T506C transition that resulted in the L169P amino acid substitution [5]. Although this variant has previously been reported, its oncogenic and hypoxia regulation function was not closely examined [19]. To study the oncogenic function of the L169P variant relative to WT *VHL*, we re-expressed the respective *VHL* gene in two common *VHL*-null human ccRCC cell lines, 786-O and RCC4. The coding sequence (CDS) of human *VHL* isoform 1 (designated as WT *VHL*) was cloned from normal human kidney tissue into a lentiviral expression vector, and the L169P was generated subsequently by the site-directed mutagenesis of the WT *VHL* construct. Sanger DNA sequencing verified the designated sequence for both constructs, as shown in Figure 1A. Upon lentiviral-mediated stable transduction, a comparable and robust expression of WT and L169P *VHL* mRNA (Figure 1B) and protein (pVHL) was achieved in the 786-O and RCC4 cells (Figure 1C,D).

Next, we evaluated the stability of L169P pVHL. The degradation of pVHL was assessed upon the addition of cycloheximide (CHX) [20], which halts the synthesis of new protein. The pVHL level was examined at 0, 3 and 6h post CHX treatment. As shown in Figure 1E, the decrement over time of WT and L169P pVHL was very comparable in the 786-O cells. This finding supports the unaltered protein stability of the L169P variant.

### 2.2. The Hypoxic Regulation of L169P VHL Is Comparable to WT VHL

We examined the hypoxic regulatory response of L169P *VHL*, as the hypoxia pathway is implicated in ccRCC oncogenesis. The protein level of *HIF-1α* and *HIF-2α*, the two major targets of *VHL*-mediated ubiquitin ligation and degradation, were examined in the parental *VHL*-null parental cells 786-O and RCC4 and their WT or L169P *VHL*-expressing derivative cells (Figure 2A,B). In the absence of *VHL* expression, a clear accumulation of *HIF-1α* and *HIF-2α* protein was observed (Figure 2A,B). Upon the reintroduction of WT or L169P *VHL*, the level of both targets was reduced to a similar extent (Figure 2A,B). These findings indicate that the L169P variant is capable of degrading both *HIF-1α* and *HIF-2α* as efficiently as its wildtype counterpart. 

We examined the broader cellular hypoxia program further, beyond the regulation of alpha subunits of *HIF*. To assess possible alterations in cellular hypoxic tension, bulk mRNA sequencing was implemented to obtain the comprehensive genetic expression profiles of the 786-O, 786-O/*VHL* L169P and 786-O/*VHL* WT cells. Using the hypoxic signature panel developed by Ragnum [21], Buffa [22] and Winter [23], the hypoxia scores for each cell line were calculated and are shown in Figure 2C. The whole list of each gene panel and their scores are shown in Figure 3. A higher hypoxia score reflects a more hypoxic state of the cells. As shown in Figure 2C, both L169P and WT *VHL* manifested similar hypoxia tensions and were both less hypoxic compared to their parental cell lines.

### 2.3. Restored Expression of WT or L169P VHL Slowed Cancer Cell Growth

Next, we evaluated the impact of L169P *VHL* on the tumorigenic phenotype. The expression of either *VHL* L169P or WT *VHL* did not alter the cellular morphology in comparison to the parental 786-O cells (Figure 4A). However, the restored expression of either *VHL*-L169P or WT *VHL* significantly suppressed the growth rate of the parental 786-O cells (Figure 4B). Meanwhile, no distinguishable difference in the growth rate was observed between the two *VHL*-expressing cells (Figure 4B). We further assessed the in vivo growth potential of these three tumor cell xenografts established on a duck CAM (chorioallantoic membrane) model. We have previously reported that a rich CAM vasculature effectively supports the growth and aggressive behaviors of numerous tumor xenografts, recapitulating their behaviors observed in mouse models [24,25]. Corroborating the in vitro results, the 786-O cells with a restored expression of either WT or L169P *VHL* exhibited slower growth rates in vivo, resulting in a significantly lower terminal tumor size and weight compared to the parental 786-O xenografts (Figure 4C,D). However, the difference in the tumor weight between the two *VHL*-expressing groups was insignificant (Figure 4D).

In this study, we analyzed the biochemical and tumorigenic activities of the L169P *VHL* variant, including protein stability, the degradation of alpha subunits of HIF, hypoxic tension and tumor-growth-suppressive ability. Overall, the L169P variant functions at a level that is comparable to WT *VHL*. 

## 3. Discussion

The functional sequela of missense mutations is more difficult to predict compared to nonsense or frameshift mutations that result in the complete loss of *VHL* protein. The crystal structure of *VHL* in the VCB complex has previously [8] been determined (https://www.rcsb.org (accessed on 1 June 2023)). Several computational algorithms have been developed that can predict the stability and function of *VHL* missense variants. Publicly accessible algorithms include I-mutant 2.0 [26], Site Directed Mutator (SDM) [27], and Crescendo [28]. Both I-mutant 2.0 and SDM calculate ddG to evaluate the thermodynamic changes in an amino acid substitution [26,27]. In these algorithms, a ddG value greater than 2 is considered “disease associated”, while one less than 2 is considered “neutral”. As shown in Table 1, the ddG values for L169P were −0.03 and −1.71, indicating it is neutral. In contrast, the CRESCENDO program assigns the L169P variant as disease-associated. This program evaluates the impact of an amino acid change by interrogating how conserved the affected region is in the protein–protein interaction. The leucine 169 residue, located in exon 3 of *VHL*, was postulated to interact with Elongin B due to its close proximity to arginine 167, which was reported to have lower regulatory effects on HIF2α [4].

Due to the equivocal designation of L169P *VHL* to “disease association” by in silico algorithms, we decided to pursue direct biochemical and functional experiments that reintroduced the *VHL* gene into two human *VHL*-null ccRCC models. In the 786-O cells, the protein stability of L169P and its capability to degrade HIFs were very comparable to WT pVHL (Figure 1D and Figure 2A,B). Our findings regarding L169P *VHL* corroborates a prior study by Rechsteiner et al. [4] that used an *HIF*αs-GFP degradation reporter assay in MEF (mouse embryonic fibroblast) cells to assess the E3 ligase function of missense variants of *VHL*, identified in ccRCC tumors. The L169P variant was amongst the 29 variants studied, and it was reported to function as WT pVHL in terms of degrading HIF1α, while the degradation of HIF2α was slightly reduced [4]. In the supplementary findings, they assigned the L169P variant as a passenger mutation, likely not pathogenic [4]. 

In this study, we pursued two additional assays besides those previously reported [4] to further clarify the potential oncogenic role of L169P *VHL* in hypoxia regulation and in tumor growth suppression. The first assay utilized mRNA sequencing and gene expression profiling to assess pVHL’s functional impact on cellular hypoxia. Bhandari et al. [29,30] reported several hypoxia score algorithms based on sets of known hypoxia marker genes. In comparison to the parental *VHL*-null 786-O cells, both the WT or L169P *VHL*-expressing cells showed a lower score by three diffent sets of hypoxia signatures (Figure 2C and Figure 3), indicating that the regulation of *HIFα*-governed hypoxia pathways were intact and were comparable under the WT or L169P *VHL*-controlled cells. The WT *VHL* gene is known to induce growth arrest by restoring cell entry into G0 or the quiescent state and to suppress tumor growth [31,32]. The impact of L169P *VHL* on tumor growth suppression was again comparable to WT *VHL* in slowing the growth of *VHL*-null 786-O cells in cultures (Figure 4B) as well as in xenografts in a CAM model (Figure 4C,D).

In a recent study, we reported that spontaneous metastasis in ccRCC models requires the presence of and cross communication between two distinct tumor cell populations, one that does not express any pVHL (*VHL*-Neg) and one that expresses WT pVHL (*VHL*-Pos) [5]. This cooperative metastasis model demonstrated that the *VHL*-Neg cells act as the metastatic driver, while the *VHL*-Pos cells play the accepting effector role [5]. In this cooperative metastatic scenario, L169P functions the same as WT *VHL*-expressing cells to achieve distant metastasis to the lungs [5]. 

The current study serves to caution that the oncogenic function of missense *VHL* variants could be difficult to determine, especially for those variants involving a single amino acid substitution in exon 3 that do not obviously reduce *VHL* protein stability. As previously reported, 30–50% of this subtle class of variants are classified as passenger, non-pathogenic variants [4]. The L169P variant belongs to this class of subtle missense variants that require thorough comprehensive biological function analyses, like those outlined here.

Many factors contribute to the difficulty in determining the oncogenic role of *VHL* variants in ccRCC, such as the complex multiple normal functions of *VHL*, its yet undetermined specific mechanism of oncogenesis, tumor heterogeneity and the presence of multiple oncogenic driver gene mutations in ccRCC. The association of VHL gene mutations with VHL syndrome and ccRCC has been known for over three decades [33]. The precise oncogenic mechanism of *VHL* remains elusive. The sophisticated conditional knockout of the *VHL* gene in the proximal tubules of mice resulted in cystic dysplasia but not renal tumors [34]. The deletion of *VHL* in conjunction with other oncogenic drivers such as *Rb1* and *Trp53* was able to generate renal tumors but unable to recapitulate the aggressive metastatic disease [35]. In the context of other genetic backgrounds, the L169P variant could possibly synergize with other oncogenic gene mutation(s), such as *BAP1* and *p53* uncovered in the highly aggressive and metastatic case of ccRCC [5], to drive tumorigenesis. However, based on the multiple lines of evidence, such as protein stability, hypoxia regulation and tumor growth suppression, we conclude at this juncture that L169P *VHL* functions comparably to WT *VHL*. Thus, the L169P variant of *VHL* alone is unlikely to drive the oncogenesis of sporadic ccRCC.

## 4. Materials and Methods

### 4.1. The Cell Culture, Reagents and Generation of Genetic Engineered Cell Lines

The 786-O and RCC4 cell lines were kindly gifted from Dr. Brian M. Shuch at the University of California, Los Angeles (UCLA). Both cells were cultured in RPMI-1640 media supplemented with 10% fetal bovine serum and 1x penicillin-streptomycin at 37 °C and in 5% CO_2_ humidified cell incubators. The coding sequence (CDS) of wildtype human *VHL* was amplified by PCR with EcoRV restriction enzyme sites on both ends, where the forward primer was 5′-AAACGGATATCATGCCCCGGAGGGCG-3′ and the reverse primer was 5′-CGATCGATATCATCTCCCATCCGTTGATGTGC-3′. The template was normal tissue cDNA from a healthy donor. The L169P variant of human *VHL* was amplified by bridging PCR in combination with primers for wildtype *VHL*. The first pair of primers were forward: 5′-AAACGGATATCATGCCCCGGAGGGCG-3′ and reverse: 5′-CAGGCTTGACTGGGCTCCG-3′. The second pair of primers were forward: 5′-CGGAGCCCAGTCAAGCCTG-3′ and reverse: 5′-CGATCGATATCATCTCCCATCCGTTGATGTGC-3′. The template was wildtype human *VHL* CDS, as mentioned above. The backbone for both *VHL*-wildtype and *VHL*-L169P overexpression plasmids was pLV-CMV-LoxP-DsRed-LoxP-eGFP (Cat#65726, Addgene, Watertown, MA, USA) without DsRed and eGFP sequences.

### 4.2. qRT-PCR, Western Blot, Cell Proliferation Assay, Chorioallantoic Membrane (CAM) Xenograft Model and VHL Stability Analysis

qRT-PCR and Western blot analyses were implemented as described previously [36]. The primers for *hVHL* were forward: 5′-GGAGCCTAGTCAAGCCTGAGA-3′ and reverse: 5′-CATCCGTTGATGTGCAATGCG-3′. The amplicon was 135 bp, located in the last exon. Western blot analysis was implemented as described previously [36]. The information of the antibodies was: anti-*VHL* (1:1000 dilution, Cat#ab135576, Abcam, Boston, MA, USA), anti-HIF1A (1:1000 dilution, Cat#ab228649, Abcam, Boston, MA, USA), anti-HIF2A (1:1000 dilution, Cat#ab199, Abcam, Boston, MA, USA), and anti-β-actin (1:1000 dilution, Cat#sc-47778, Santa Cruz Biotechnology, Dallas, TX, USA), and the secondary antibodies were goat-anti-rabbit (1:1000 dilution, Cat#111-035-045, Jackson ImmunoResearch Laboratories, West Grove, PA, USA) and goat-anti-mouse (1:1000 dilution, Cat#115-035-062, Jackson ImmunoResearch Laboratories, West Grove, PA, USA).

A cell proliferation assay was carried out with MTS reagent from Promega (Cat#G3580, Madison, WI, USA). On day 0, log phase cells were seeded in a flat-bottom 96-well plate at 1000 cells/well. The absorbance of each well in all groups was examined every 24 h by a Multiskan MK3 microplate reader (Thermofisher, Waltham, MA, USA) at 490 nm.

A chorioallantoic membrane (CAM) cancer cell xenograft model was established by implanting the cancer cells on top of the highly vascularized CAM, upon which the cancer cells invaded into the duck embryonic blood cells to obtain nourishment. The cancer cells were implanted on the embryo at day 15 upon the delivery of fertilized duck embryos and were allowed to grow until day 28. Tumors were harvested and weighted for statistical analysis [24,25,37].

For the *VHL* stability analysis, 786-O cells were cultured and seeded in a 6-well plate at 5 × 10^5^ cells per well on day 0. On day 1, each well was transfected with either 3000 ng of wildtype *VHL* or L169P variant plasmid with 9 μL of Promega FuGENE transfection reagent (Cat#E2311, Promega, Madison, WI, USA) followed by 24 h of incubation. Upon harvesting, cycloheximide (CHX, 100 μg/mL, Cat#357420010, Acros Organics, Waltham, MA, USA) was added to each well, and RIPA lysis buffer was used to harvest the cells at 0, 3 and 6 h upon CHX addition. The harvested protein was subjected to Western blot analysis, as mentioned earlier.

### 4.3. Bioinformatic Analysis and Hypoxic Score Calculation

The mRNA from the 786-O parental cell line as well as the derived cells overexpressed with wildtype or L169P human *VHL* was extracted and sequenced for mRNA expression profiles at UCLA Technology Center for Genomics & Bioinformatics (TCGB) core. 

Libraries were prepared with a KAPA Stranded mRNA-Seq kit. The workflow consisted of mRNA enrichment and fragmentation, first-strand cDNA synthesis using random priming followed by second-strand synthesis converting cDNA:RNA hybrid to double-stranded cDNA (dscDNA), and it incorporated dUTP into the second cDNA strand. cDNA generation was followed by end repairing to generate blunt ends, A-tailing, adaptor ligation and PCR amplification. Different adaptors were used for multiplexing samples in one lane. Sequencing was performed on an Illumina NovaSeq 6000 device for a PE 2 × 100 run. Data quality checking was carried out on an Illumina SAV device. Demultiplexing was performed with the Illumina Bcl2fastq v2.19.1.403 software.

The Partek flow software (Partek^®^ Flow^®^ software, version 7.0 Copyright©; 2019 Partek Inc., St. Louis, MO, USA) was used for the data analysis. The alignment and generation of the raw read counts per gene were performed using STAR-2.7.9a [38,39] with human reference genome GRCh38 genome and were annotated with Ensembl 104 GTF. Normalization of transcript counts was performed using CPM (counts per million) normalization followed by adding 0.0001. The Partek GSA algorithm was used for differential gene expression, with *p* < 0.05 and FDR < 0.05, and 2Fold was used for cutoffs to obtain lists of significantly expressed genes from the sample comparisons.

The hypoxia scores of the three cell lines were calculated using the mRNA-based signatures developed by Ragnum [21], Buffa [22] and Winter [23]. The raw read counts from the three cell lines were normalized with the TCGA database level 1 data using the DESeq2 package v1.38.3 accessed by 1 June 2023, and the hypoxia scores were calculated as described in Bhandari et al. [29].

### 4.4. Statistical Analysis

All experiments were repeated three times, and data are presented as means ± SEM. One-way ANOVA was used for the three group comparisons between the parental cell line, *VHL*-WT- and *VHL*-L169P-overexpressed cells followed by planned post hoc comparisons.

## Figures and Tables

**Figure 1 ijms-24-14075-f001:**
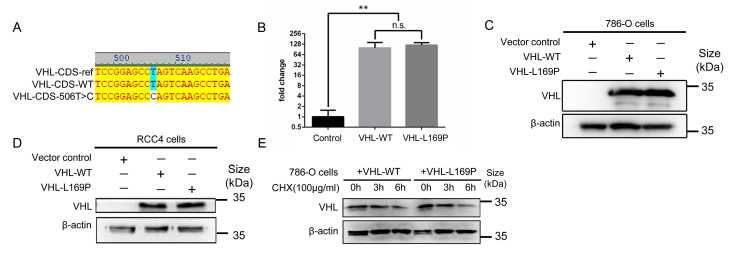
Protein stability of L169P variant is comparable to WT pVHL. (**A**) Sequence alignment of the Sanger sequencing results from plasmids encoding wildtype *VHL* or c.506T>C point mutated *VHL* and the reference sequence, as made by the Vector NTI software v8.0. (**B**) qRT-PCR analysis of *VHL* in 786-O cells vector control, VHL/WT and VHL/L169P. (**C**,**D**) Western blots identifying *VHL* protein upon reintroduction into the parental cell line 786-O (**C**) and RCC4 (**D**). (**E**) A total of 100 μg/mL CHX was applied to evaluate the *VHL* degradation rate at time points of 0, 3 and 6 h after mRNA translation was blocked in 786-O cells. (**: *p* < 0.01, n.s.: no statistical significance).

**Figure 2 ijms-24-14075-f002:**
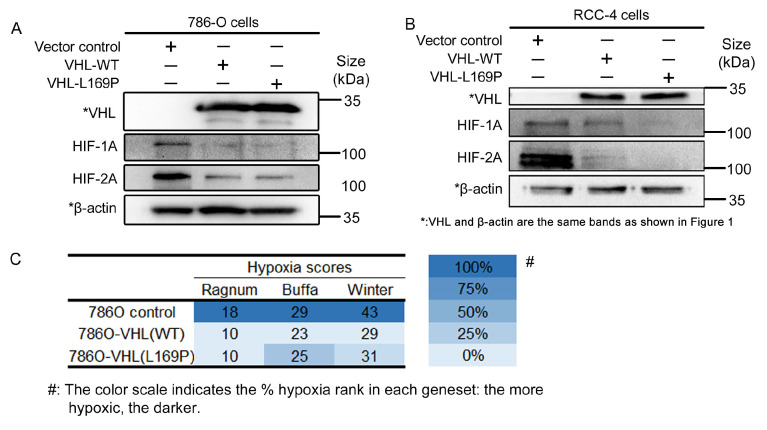
L169P variant showed a similar capability to WT pVHL in decreasing hypoxic tension by degrading HIF-1A and HIF-2A. (**A**) Western blot analysis of HIF1A and HIF2A upon wildtype and L169P *VHL* overexpression in 786-O cells. (**B**) Western blot analysis of HIF1A and HIF2A upon wildtype and L169P *VHL* overexpression in RCC4 cells. (**C**) Hypoxic scores calculated by the gene signature panels from Ragnum et al. [21], Buffa et al. [22] and Winter et al. [23] on the parental 786-O cells as well as the derived cell lines that were either wildtype or L169 overexpressed.

**Figure 3 ijms-24-14075-f003:**
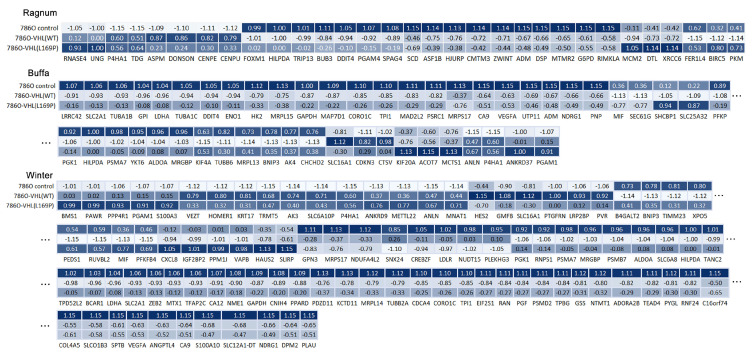
Gene list and heatmap from the hypoxic signature panels from Ragnum et al. [21], Buffa et al. [22] and Winter et al. [23]. The darkness of the color indicates the relative expression level for each gene labeled on the bottom. The darker, the higher expression level of gene in that cell.

**Figure 4 ijms-24-14075-f004:**
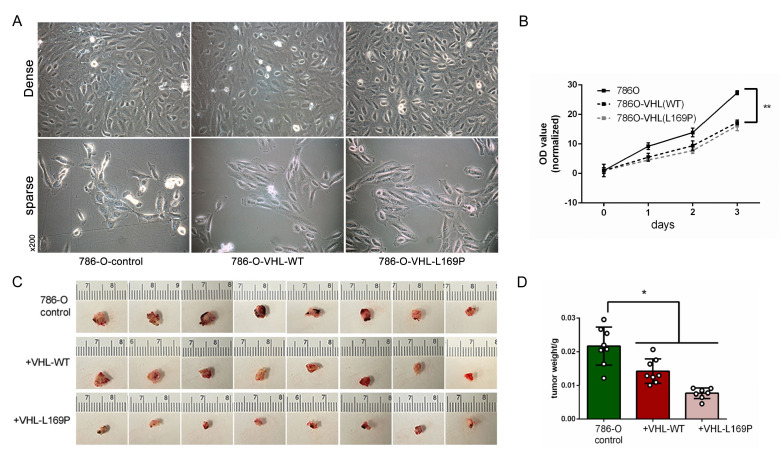
Overexpression of either wildtype or L169P *VHL* slowed cell growth both in vitro and in vivo. (**A**) Phase contrast images of the 786-O parental cells as well as the overexpressed cell lines with either wildtype or L169P *VHL* in both dense and sparse cultured modes (×200 magnification field). (**B**) Cell proliferation was analyzed by an MTS assay among the three cells for up to three days. (**C**) Gross images of the primary tumors formed in the duck CAM model from the three cell lines after 13 days post implant. (**D**) Statistical analysis of the tumor weight from (**C**). (*: *p* < 0.05, **: *p* < 0.01).

**Table 1 ijms-24-14075-t001:** Human VHL in silico algorithm analysis results. * indicates algorithms were accessed and results were updated by 1 June 2023.

Gene Variant	I-Mutant 2.0 [26] *	Site Directed Mutator (SDM) [27] *	Crescendo [28] *
h*VHL*, L169P	−0.30	−1.71	Conserved, likely pathogenic

## Data Availability

The RNAseq data are available in NCBI SRA with project ID#PRJNA763575, https://www.ncbi.nlm.nih.gov/bioproject/PRJNA763575.

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
