# Peer review of "VHL L169P Variant Does Not Alter Cellular Hypoxia Tension in Clear Cell Renal Cell Carcinoma"

_ijms, 2023, doi:10.3390/ijms241814075_

Round 1

Reviewer 1 Report

The manuscript presented for review is interesting, however, the authors did not avoid errors that should be corrected before potential publication:

1. verse 19 - gene expression program??

2. verse 26 - according to the reviewer, the sentence is redundant

3. names of genes should be written in italics - to be corrected throughout the paper

4. In the opinion of the reviewer, creating a separate chapter for figures, tables etc. is unnecessary, they should rather be included in the text of the manuscript where their descriptions appear in the text

5. Figures 2C and 2D should be separate figure then they would be more readable especially 2D which is now microscopic

6. figure 1A - how was it created? Is it your own work or from some program? if it's from some program it should have a citation and access date.

7. why was the CHX effect assessed only on one cell line? why and on what basis was 100 ug/ml CHX selected?

8. Figure 3A - there is no spectacular difference between the control and variants?

9. caption under figure 3 - verse 164 - shouldn't it be 3 days?

10. what was the purpose of the MTS test? whether to assess the effect of the variant on cell survival? or CHX ?

11. did the authors take into account the assessment of the mRNA level of the gene depending on the variant? and additionally also to assess the relationship of this level of mRNA to protein?

12. Discussion section - lines 167-180 - in the reviewer's opinion, this fragment should be included in the results section as the results of bioinformatics analysis; additionally, no description of these programs in the Materials and Methods section and no access dates

13. In the discussion section, too many repetitions of the description of the results and not enough considerations about the results of own and other researchers or references to the clinical significance of this change

14. verses 70-83 - not fully understood the sense of including this information in the publication since the researched material were rented cell lines? Or maybe the authors themselves brought them out? if so, there is no consent number of the bioethics committee in this case

15. do the authors have the consent of the authors of the scheme included in the suplemmentary material for its publication in this article?

Reviewer 2 Report

The study of the features of the development of light cell kidney cancer is a significant event. It is known that in most cases this is due to the functional inferiority of the VHL protein. It is worth noting that the sequencing of this gene is a significant scientific event. It is clear that genetic changes are not always associated with the loss of protein function. Such works undoubtedly expand the fundamental understanding of cancer genetics. However, the role of these studies in practical medicine is practically absent in the work. How significant is the study of somatic and germinative changes in the VHL gene, still remains poorly understood.

The study of the features of the development of light cell kidney cancer is a significant event. It is known that in most cases this is due to the functional inferiority of the VHL protein. It is worth noting that the sequencing of this gene is a significant scientific event. It is clear that genetic changes are not always associated with the loss of protein function. Such works undoubtedly expand the fundamental understanding of cancer genetics. However, the role of these studies in practical medicine is practically absent in the work. How significant is the study of somatic and germinative changes in the VHL gene, still remains poorly understood.

Quality of English - good

Round 2

Reviewer 1 Report

The author include and discussed all reviewer suggest and question that is why in reviewer opinion this manuscript should be published in this form.